# Deep Sequencing of Early T Stage Colorectal Cancers Reveals Disruption of Homologous Recombination Repair in Microsatellite Stable Tumours with High Mutational Burdens

**DOI:** 10.3390/cancers14122933

**Published:** 2022-06-14

**Authors:** Jun Li, Pascal Steffen, Benita C. Y. Tse, Mahsa S. Ahadi, Anthony J. Gill, Alexander F. Engel, Mark P. Molloy

**Affiliations:** 1Bowel Cancer and Biomarker Laboratory, Kolling Institute, School of Medical Sciences, Faculty of Medicine and Health, The University of Sydney, Sydney, NSW 2006, Australia; jun.li2@sydney.edu.au (J.L.); pascal.steffen@sydney.edu.au (P.S.); benita.tse@sydney.edu.au (B.C.Y.T.); 2NSW Health Pathology, Department of Anatomical Pathology, Royal North Shore Hospital, Sydney, NSW 2065, Australia; mahsa.seyedahadi@health.nsw.gov.au (M.S.A.); anthony.gill@health.nsw.gov.au (A.J.G.); 3Sydney Medical School, Faculty of Medicine and Health, The University of Sydney, Sydney, NSW 2006, Australia; alexander.engel@sydney.edu.au; 4Colorectal Surgical Unit, Royal North Shore Hospital, Sydney, NSW 2065, Australia

**Keywords:** colorectal cancer, tumour mutation burden, homologous recombination repair, *ATM*

## Abstract

**Simple Summary:**

Stage IIIA are rare early T stage colorectal cancers that are not widely reported in genome sequencing studies. We aimed to investigate genomics features in common cancer genes associated with lymph node metastasis from these tumours. We discovered that microsatellite stable tumours with high mutational burdens showed different genomic variant frequencies compared to tumours with low mutation burdens. The high burden microsatellite stable tumours showed co-occurrence of mutations in homologous recombination DNA repair genes that where not observed in the microsatellite stable cancers with low mutational burdens. This finding highlights a potential therapeutic approach for high mutation burden microsatellite stable colorectal cancers given the successful clinical implementation of drugs blocking homologous recombination DNA repair in other cancers.

**Abstract:**

Early T stage colorectal cancers (CRC) that invade lymph nodes (Stage IIIA) are rare and greatly under-represented in large-scale genomic mapping projects. We retrieved 10 Stage IIIA CRC cases, matched these to 16 Stage 1 CRC cases (T1 depth without lymph node metastasis) and carried out deep sequencing of 409 genes using the IonTorrent system. Tumour mutational burdens (TMB) ranged from 2.4 to 77.2/Mb sequenced. No stage-related mutational differences were observed, consistent with reanalysis of The Cancer Genome Atlas (TCGA) Stage I and IIIA datasets. We next examined mutational burdens and observed that the top five cancers were microsatellite stable (MSS) genotypes (mean TMB 49.3/Mb), while the other 16 MSS cancers had a mean TMB of 5.9/Mb. To facilitate comparison with TCGA hypermutator CRC, we included four microsatellite instability-high (MSI-H) samples with the high mutation burden MSS cases to form a TMB-High group. Comparison of TMB-High with TMB-Low groups revealed differences in mutational frequency of *ATM*, *ALK*, *NSD1*, *UBR5*, *BCL9*, *CARD11*, *KDM5C*, *MN1*, *PTPRT* and *PIK3CA*, with *ATM* and *UBR5* validated in reanalysis of TCGA hypermutator Stages I and IIIA samples. Variants in *ATM* were restricted to the TMB-High group, and in four of five MSS specimens, we observed the co-occurrence of mutations in homologous recombination repair (HRR) genes in either two of *ATM*, *CDK12*, *PTEN* or *ATR*, with at least one of these being a likely pathogenic truncating mutation. No MSI-H specimens carried nonsense mutations in HRR genes. These findings add to our knowledge of early T stage CRC and highlight a potential therapeutic vulnerability in the HRR pathway of TMB-H MSS CRC.

## 1. Introduction

Colorectal cancer (CRC) maintains a high worldwide disease burden as the third most common malignancy and the second leading cause of cancer mortality [1]. In 2020, CRC accounts for 10% of global cancer incidence and 9.4% of cancer deaths [2]. Molecular studies have revealed two major and distinct genetic/epigenetic pathways for sporadic CRC development [3]. The conventional adenoma–carcinoma pathway is driven by APC/beta-catenin/Wnt disruption, often coupled with *KRAS* oncogene activation resulting in chromosomal instability. CRC arising in this pathway is common and these tumours have proficient DNA mismatch repair mechanisms and are characterised as microsatellite stable (MSS) [4]. In contrast, CRC arising from the serrated pathway is characterised by high levels of CpG island methylator phenotype (CIMP) due to the epigenetic silencing of the *MLH1* DNA mismatch repair gene [5]. Deficient mismatch repair and high microsatellite instability (MSI-H) leads to the accumulation of many hundreds of mutations fuelling carcinogenesis. Large-scale genomic profiling projects such as The Cancer Genome Atlas (TCGA) have contributed significantly more details as to the identity and frequencies of driver and other mutations in CRC [6]. For example, review of TCGA-COAD dataset revealed *APC* (77%), *TP53* (58%), *KRAS* (44%), *MUC16* (38%), *PIK3CA* (31%) and *OBSCN* (30%) as the top six mutated genes in sporadic colon adenocarcinoma (https://portal.gdc.cancer.gov/projects/TCGA-COAD, (accessed on 30 November 2021).

However, not all AJCC/TNM stages of CRC [7] are equally represented in TCGA data. One of the most under-represented is Stage IIIA, diagnosed in approximately 10% of all locally advanced CRCs (LAC), and signifies the earliest stage of LAC. The combined TCGA-COAD and TCGA-READ datasets consist of only 12 patients with Stage IIIA CRC, greatly limiting our knowledge on the molecular drivers underlying tumour development. Stage IIIA CRC (T1-T2 N1/N1c M0 or T1 N2a M0) is defined by lymph node metastasis (LNM), despite the primary lesion having a limited invasive depth no greater than into the muscularis propria (T1-T2). Stage IIIA patients have good prognosis, although a considerably higher metastatic relapse rate compared with Stage I tumours [8,9] which show similar invasive depth, but no lymph node metastasis (T1-2 N0 M0). Determining the likelihood of LNM in T1-T2 cancers remains an important unmet clinical need, especially from endoscopic polypectomy specimens. This is because many patients with a T1-T2 diagnosis post polypectomy undergo additional radical surgery, where the subsequent diagnosis of LNM is reported to be as low as 8–16%, and these patients are exposed to potential postoperative complications [10,11]. Using proteomics, we have previously shown that Stage IIIA CRCs are enriched in protein regulators of the epithelial–mesenchymal transition (EMT) and proposed that this may be useful in identifying high-risk polypectomy specimens [12]. To complement this study, molecular genomic analysis focusing on Stage IIIA CRC is needed to establish mutational profiles, the risk of LNM and further improve prognostication and treatment approaches.

Stage IIIA CRC is infrequent. We undertook massively parallel sequencing on our Stage IIIA CRCs to establish their common mutational profiles and compared this to a matched Stage I CRC cohort. By focusing on early T stage CRCs, we made interesting observations on mutational burden and pathway defects that have not been widely reported.

## 2. Materials and Methods

### 2.1. Specimens

The study was approved by the Northern Sydney Local Health District Human Research Ethics Committee (RESP/18/248). FFPE colorectal adenocarcinoma specimens from patients who underwent surgical resection between 2007 and 2018 were acquired from the Department of Anatomical Pathology, Royal North Shore Hospital (Sydney, NSW, Australia). Lynch syndrome cases were excluded from the study. We utilised specimens where sufficiently high-quality DNA could be obtained from formalin-fixed paraffin-embedded (FFPE) blocks, resulting in 16 Stage I and 10 Stage IIIA cases based on their synoptic pathological staging according to the 8th edition AJCC/TNM staging system [7]. Mismatch repair protein (MMRp), immunoexpression (MSH2, MSH6, MLH1, PMS2) and MSI status were examined as part of routine clinical histopathology reporting, showing four cases were MSI-H cancers. Combined BRAFV600E/MMRp status was determined and showed two MSS/BRAF mutant cases (2/26), four MSI-H/BRAF mutant cases (4/26), and twenty MSS/BRAF wild-type cases (20/26).

### 2.2. Next Generation Sequencing

Hematoxylin and eosin-stained tissue sections from FFPE tumour samples first were reviewed by a qualified pathologist (M.A.) and the tumour region was marked out on the slides. Extraction of tumour area was achieved by macrodissection. DNA/RNA recovery followed the manufacturer’s protocol of AllPrep DNA/RNA FFPE Kit (QIAGEN, Hilden, Germany).

### 2.3. Ion AmpliSeq™ Comprehensive Cancer Panel

Targeted sequencing was achieved using Ion AmpliSeq™ Comprehensive Cancer Panel (ThermoFisher Scientific, Waltham, MA, USA), which consisted of 409 key cancer-related genes. Library preparation for each specimen was performed using the Ion Ampliseq Library Kit 2.0 (ThermoFisher Scientific) according to the manufacturer’s instructions. The prepared libraries were partially digested and phosphorylated using the FuPa reagent, ligated to different barcode adapters using the Ion Xpress Barcode Adapters Kit (ThermoFisher Scientific), then purified. The purified libraries were quantified using the Ion Library TaqMan Quantitation Kit (ThermoFisher Scientific).

Data analyses for variant calling (SNVs/multinucleotide variants (MNVs) and indels) and annotation were performed in Ion Reporter software version 5.14 (ThermoFisher Scientific) following tumour-only analysis workflows of Oncomine Tumour Mutation Load—w3.1—DNA and the Oncomine Variant Annotator v3.0 plug-in (ThermoFisher Scientific) using default settings. We applied the Oncomine Extended (version 5.12) filter chain to identify somatic variants by interrogating public datasets to filter any possible germline SNVs. Human genome build 19 was used as reference in alignment. A minimum sequencing depth of 500× was considered as adequate sequencing depth, and variant allelic fraction (VAF) of 5% was used as a cut-off for positive variants. Gene/variant summaries and plots were prepared by maftools [13]. Comparisons of mutation gene frequencies were carried out with Fisher’s exact test with p-value corrections. Sample size calculations for Fisher’s exact test were performed using G*Power 3.1, available to download from https://www.psychologie.hhu.de/arbeitsgruppen/allgemeine-psychologie-und-arbeitspsychologie/gpower.html, (accessed on 19 April 2022).

### 2.4. Retrieval of TCGA datasets

TCGA COAD and READ projects were used as an independent validation data source. COAD and READ data were downloaded from GDC data portal (https://portal.gdc.cancer.gov/, (accessed on 30 November 2021) and inclusive criterion were disease type of “adenomas and adenocarcinomas” and AJCC pathologic stages of “stage I” and “stage IIIA”. In total 96 Stage Is and 12 Stage IIIAs were available.

## 3. Results

### 3.1. Robust Deep Targeted Sequencing of Early T Stage Colorectal Cancer

TCGA COAD and READ datasets report molecular genomics on 550 cases, with only 12 cases defined as Stage IIIA CRC. Over the past decade in our institution, Stage IIIA CRC was diagnosed in approximately 4% of all surgical cases. We identified 10 Stage IIIA cases where FFPE blocks were available, and DNA was recovered with sufficient quality for deep sequencing. A randomly selected cohort of 16 Stage I CRCs matched for gender and tumour site with sufficiently high-quality DNA extracted from FFPE was used for comparison as early T stage tumours without lymph node metastasis. Clinicopathological characteristics of the cohort used in this study is shown in Table 1. Stage IIIA patients were younger and by definition had nodal involvement or subserosal deposits as the only significant variables in the cohort.

DNA from 26 FFPE CRC specimens were analysed using the IonTorrent AmpliSeq™ Comprehensive Cancer Panel, which targeted the exons of 409 tumour suppressor genes and oncogenes. One sample was excluded from data analysis, as the sequencing was below the quality threshold. For the remaining 25 samples, the range of mean sequencing depth was 876–2253-fold; mapped reads were 15,073,849–35,403,831; uniformity was between 86.59 and 96.25%; on-target reads were 93.66–99.19%. The dbSNP concordance rate was 0.836–0.992. These metrics confirm robust deep sequencing data quality.

Tumour mutation burden (TMB) varied from 2.4 per Mb to 77.2 per Mb, mean 16.5 per Mb. The top five cancers by TMB were MSS genotypes (no evidence for MMR deficiency by IHC) with mean TMB 49.3/Mb. There was a clear distinction between these TMB-High MSS cancers and other MSS cancers which we classified as TMB-Low (mean 5.9/Mb) (Figure 1). Four MSI-H cancers were included in our dataset, and as these showed a mean TMB of 17.6/Mb, we grouped these with the TMB-H MSS cancers. This produced two specimen groups with statistically significant differences in mutational burdens (Mean TMB-H group 35.2/Mb, Mean TMB-Low group 5.9/Mb; *t*-test *p* < 0.001). This grouping strategy was useful to enable comparisons with TCGA hypermutator and nonhypermutator Stage I and IIIA specimens.

The ultrahypermutator gene *POLE* is associated with a TMB > 100/Mb, as determined from the analysis of over 78,000 adult cancers, where 75% of these tumours were MSS [14]. Consistent with our findings, MSI-H TMB cases were restricted to a 10–100/Mb range. Lee et al. has also shown that pathogenic mutations in homologous recombination repair (HRR) genes were associated with the MSS hypermutated phenotype without POLE exonuclease domain mutations [15]. We note that *POLE* is not included in the AmpliSeq panel, so we could not definitely exclude this as a driver mutation in the MSS high tumour burden samples. Nonetheless, given all our cases were below the >100/Mb TMB load commonly associated with the *POLE* ultrahypermutator phenotype and the low 1–2% reported CRC incidence [16], there is low likelihood this phenotype is included in our sample cohort.

### 3.2. No Stage-Related Mutational Differences between Stage IIIA vs. Stage I in TMB-Low Group

To further elucidate the molecular differences between Stage IIIA and Stage I CRC, we focused on the TMB-L group, consisting of eight Stage I and eight Stage IIIA samples. In the TMB-H group, there were insufficient Stage IIIA cases for a meaningful analysis. In comparing all 409 cancer-related genes, *KRAS* (62.5% vs. 12.5% in Stage IIIA vs. Stage I) and *TP53* (85.7% vs. 50%) trended to a higher mutation frequency in Stage IIIA compared to Stage I cancers, although this did not reach statistical significance (Table 2). No stage-related difference was observed in the mutational frequencies of *APC*, *PIK3CA*, *FBXW7* and *SMAD4* CRC driver genes. We compared our findings with TCGA COAD and READ cohorts, adopting the TCGA-defined threshold for discriminating hypermutators from nonhypermutators (i.e., >12 per 10^6^ (total mutations > 728) [6]). As shown in Appendix A, the comparison of 69 Stage I and 12 Stage IIIA TCGA specimens [6] showed no significant difference in gene frequencies, consistent with our observations from a smaller size sample cohort.

We next used Oncoplots to compare frequent gene variants in the TMB-L cohort with the findings from the TCGA COAD and READ nonhypermutated cohort (Figure 2). Taking our top ten genes, we noted consistency in the top three genes: *APC* (88% vs. 85%), *TP53* (69% vs. 57%) and *KRAS* (38% vs. 43%), but observed higher frequencies of the remaining seven genes (*TAF1* (31% vs. 1%), *FBXW7* (31% vs. 11%), *NTRK1* (25% vs. 0%), *ARID2* (25% vs. 2%), *RET* (25% vs. 2%), *TAF1L* (25% vs. 4%) and *USP9X* (19% vs. 4%)) compared with the TCGA cohorts (Figure 2). The reciprocal analysis using the top 10 genes from TCGA cohorts is presented in Appendix A. *APC* (85%), *TP53* (57%), *KRAS* (43%), *PIK3CA* (20%), *SYNE1* (19%), *LRP1B* (12%), *FBXW7* (11%), *CARD11* (9%), *ATM* (9%) and *NRAS* (9%) are the most frequent 10 genes in TCGA COAD and READ. We detected fewer variants in *PIK3CA* (6% vs. 20%) and *ATM* (0% vs. 9%) compared with the TCGA cohorts, while the frequencies of the other genes were consistent. It is noteworthy that only 12 specimens of Stage IIIA exist in the TCGA cohort, and the small cohort sizes of our data and the TCGA cohort is a likely contributing factor to the disparate frequencies observed for some genes. Different strategies for germline filtering (previously reported SNPs vs germline filtering) and sequencing techniques (deep panel vs. WXS) further contributes to the differences between the two studies.

### 3.3. Mutational Spectrum of TMB-H and TMB-L CRC

This study allowed us to compare the TMB-H and TMB-L groups of early T stage CRC, where we observed the different frequencies of the top mutated genes. *ATM*, *ALK*, *NSD1*, *UBR5*, *BCL9*, *CARD11*, *KDM5C*, *MN1*, *PTPRT* and *PIK3CA* were common cancer genes with significant distribution differences between the TMB-H and TMB-L groups (Table 3 and Appendix A). There was sufficient statistical power to confirm *ATM* and *UBR5* as high-frequency mutated genes in early T stage TCGA hypermutator specimens. The absence of other gene associations may be a consequence of the small sample sizes of the early T stage TMB-H and TCGA hypermutated samples. Moreover, this further restricted us from determining whether the high-mutation frequency was due to an elevated background mutation rate [17]. In the TMB-L group, the most common mutated cancer genes were *APC*, *TP53* and *KRAS*, which were also commonly mutated in the TCGA nonhypermutated samples (Figure 2).

*APC* was less frequently mutated in TCGA hypermutator specimens, and this trend was repeated in our dataset, but did not reach the significance threshold. When *APC* was mutated in the TMB-H cancers, a more diverse composition of variant types was observed: three missense, three nonsense, one frameshift indels and one splice site variant. Conversely, among the 16 TMB-L samples, none of the *APC* alterations was missense/synonymous. Instead, we observed potentially high-impact deleterious *APC* mutations consisting of fourteen nonsense mutations and five frameshift indels. The lollipop plot (Figure 3) plots the distributions of the *APC* variants in the context of functional protein domains. The TMB-H CRC mutations were dispersed over the codons of 516–2620, while the TMB-L CRC contained 58% (11/19) of amino acid changes within the mutation cluster region (MCR, codons 1282–1581), which is in the central part of the *APC* coding frame involved in β-catenin downregulation [18]. The clustering of the *APC* variants (49%) also can be observed in the MCR of the TCGA COAD-READ nonhypermutators (Appendix A).

### 3.4. Deficiency of Homologous Recombination Repair in TMB-H CRC

Interestingly, the variants in *ATM* were only present in the TMB-H tumours. Significant differences of the *ATM* distribution were also observed in the analysis of the early T stage TCGA hypermutated (60%) and nonhypermutated groups (9%) (*p* = 0.0004) (Table 3). According to the TCGA publication that analysed all the stages of CRC [6], *ATM* did not appear as a top ranked recurrently mutated gene in either the hypermutated (rank 38) or nonhypermutated (rank 20) cohorts. Our dataset highlighted *ATM* as a potential indicator for TMB-H or hypermutated CRC, evident in early T stage cancers here, and consistent with the data available from TCGA early T stage CRC. That drove our attention to the homologous recombination repair (HRR) pathway, one of the most important cellular pathways for the repair of double-strand DNA breaks, which has been associated with cancer predisposition and increased sensitivity to chemotherapeutic agents that cause double-strand breaks [19]. On the AmpliSeq panel, 13 HRR pathway genes are present: *ATM*, *PTEN*, *BRIP1*, *ATR*, *PALB2*, *MRE11A*, *RAD50*, *CHEK1*, *CHEK2*, *FANCD2*, *FANCA*, *XRCC2* and *CDK12.* The Figure 4 Oncoplot shows the co-occurrence of multiple mutations in either two of *ATM*, *CDK12*, *PTEN* and *ATR* in the TMB-H MSS CRC identified here for the first time (Appendix A). In our data, four out of five TMB-H MSS CRCs carry at least two truncating mutations or multihit deleterious lesions in these four HRR genes. The remaining TMB-H MSS cases display a missense variant in *FANCD2* only, which is predicted to be pathogenic, suggesting a distinct mechanism of high-mutational burden rise. Importantly, none of the MSI-H CRC showed the HRR gene-truncating mutations.

### 3.5. Oncogenic Pathway Analysis

Since our study used targeted sequencing of key cancer genes, we evaluated frequent alterations in ten canonical signalling pathways, as described in previous TCGA reports [20] (Figure 5). We excluded variants of unknown significance by following the steps stated in the TCGA paper [20]. WNT, TP53 and RTK-RAS were the most frequently affected pathways among the early T stage cancers (Figure 5A). The TCGA COAD/READ dataset demonstrated similar findings of WNT, TP53, RTK-RAS and PI3K as the most frequently mutated pathways. High-frequency (>40% samples) affected pathways in the TMB-H cancers were TP53, RTK-RAS, PI3K, WNT and NOTCH (Figure 5B). In TMB-L cancers, the highest frequency affected pathways were WNT, TP53 and RTK-RAS (Figure 5C). The discrepancies in the MYC pathway compared with the TCGA data is due to the limited number of MYC genes in the AmpliSeq panel.

## 4. Discussion

In this study, we set out to profile common cancer-related mutations that are seen in early T stage CRCs, with a particular interest in Stage IIIA CRC given the limited genomic data available for this disease group. We adopted an in-depth massively parallel sequencing using the Ion AmpliSeq™ Comprehensive Cancer Panel, which contained 409 key cancer-related genes and contributed an additional 80% of genomic profiles for Stage IIIA CRC cases beyond that reported by TCGA. While recognising the small study size, our analysis did not reveal any differentially distributed gene mutation that would stratify LNM positive T1-2 patients from Stage I CRC patients.

The use of massively parallel sequencing enabled the calculation of TMB, enabling us to define TMB-H and TMB-L for separate analyses. A key observation arising was the enrichment of some MSS CRC in the TMB-H group (24% cases). Hypermutated and MSI-H CRC are predictive factors for favourable response to immunotherapy and improved survival [21]. Interestingly, seven out of nine hypermutated samples (78%) originated from Stage I patients, whilst a more balanced composition of eight Stage I vs eight Stage IIIA was found in the TMB-L group. Four TMB-H cases were in the right side of the colon, one from the left colon and four were rectal cancers. Hu et al. examined hypermutated CRC samples in the TCGA data and discovered a similar distribution as our data, with 77% of TCGA hypermutated CRC samples derived from Stage I/II patients and 80% located in the right side of the colon [22].

Our study points out the promising role of homologous recombination repair (HRR) pathway gene mutations, specifically the co-occurrence of *ATM* with *CDK12*, *PTEN* or *ATR* in identifying hypermutated MSS CRC; a finding we confirmed is observable in the larger TCGA-COAD/READ Stage I/IIIA dataset (Appendix A). This has potential translational significance, as HRR deficiency (HRD) is targetable with PARP inhibitors to exploit synthetic lethality, as is the case with BRCA1/2 ovarian cancer [23]. Exploiting such a vulnerability in MSS CRCs that carry “BRACness” genetics [24] would be a paradigm shift, where chemotherapy has remained the only treatment option for advanced MSS CRC for decades.

A significant barrier to targeting HRD in all cancers has been establishing best practice to identify the characteristics of HRD tumour status. In clinical practice, it is currently unfeasible to use WES or deep panel sequencing as we applied here to determine TMB and find hypermutated MSS. Identifying BRACness CRC may be feasible using a targeted panel of HRR genes, as applied in other cancer settings [25]. Lee et al. analysed hypermutated MSS CRC with *BRCA1/2* somatic truncations using all stages of CRC from TCGA PanCancer study as a discovery set (*n* = 21) and the metastatic CRC subset of MSK-IMPACT (Memorial Sloan Kettering Cancer Center [MSKCC]) as a validation cohort (*n* = 41). They concluded that somatic truncations of *BRCA1/2*, in combination with other HRR genes, can identify MSS hypermutators with or without known pathogenic exonuclease domain mutations in *POLE* [15]. Interestingly, in an analysis of the true positive rates to identify MSS hypermutant CRC, Lee et al. predicted the best performing three individual genes were *PTEN*, *ATM* and *ATR.* Since *BRCA1/2* are not included on the Ion AmpliSeq™ Comprehensive Cancer Panel, we could not confirm this in our dataset of early T stage CRC, so we examined their frequencies in TCGA data (Appendix A) and found the mutational frequencies of *BRCA2* and *BRCA1* were only 30% and 20%, respectively, in hypermutant early T stage CRC. In our study, we report the co-occurrence of truncating mutations in *ATM*, *CDK12*, *PTEN* or *ATR* is present in 80% of MSS CRC with TMB-H and absent from TMB-L MSS cancers. When applied to the TCGA data, the co-occurrence of *ATM* with another HRR gene outperformed the use of *BRAC1/2*, yielding a 60% detection rate. Discrepancy on the weight of *BRCA1/2* in predicting hypermutant MSS cases might be related to advanced disease stages dominating the report of Lee et al. [15], while our study focused on early T stage CRC.

While the sample size of our study, and that reported in the TCGA for early T stage CRC with MSS, is small, our findings provide a testable pathway for the potential exploitation of HRD in hypermutant MSS CRC. We believe this is the first report that shows deleterious *ATM* mutations significantly enriched in hypermutated early T stage CRC. A future large prospective study will be required to demonstrate the translational feasibility of our finding as promising biomarkers to identify suitable patients for treatments targeting HRD.

## 5. Conclusions

This study has significantly increased the molecular genomics knowledge base for Stage IIIA CRC. We made the interesting observation in MSS CRC with high TMB that these tumours are defined by nonsynonymous variants affecting genes of the HRR pathway, commonly *ATM*, but also others. Moreover, these MSS tumours harboured co-occurring mutations in HRR genes, including at least one truncating mutation. This finding may be significant if observed in advanced CRC, as it highlights a potential vulnerability which may be exploited therapeutically using the combination of agents that cause DNA single strand breaks and PARP inhibitors.

## Figures and Tables

**Figure 1 cancers-14-02933-f001:**
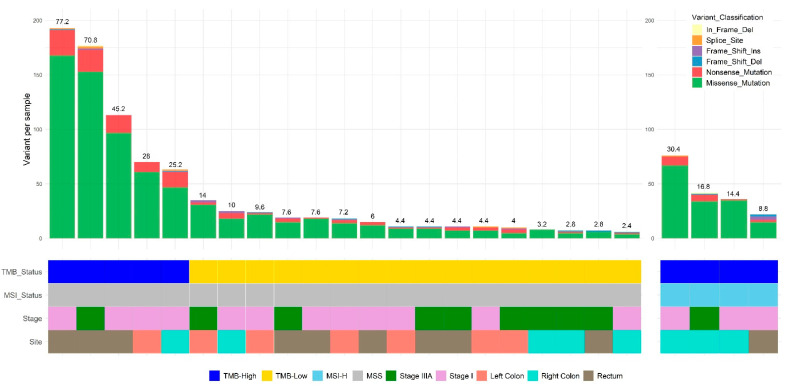
Summaries of variant profiling in TMB-High and TMB-Low Stage I and IIIA CRC. Numbers above bar chart denote TMB values of each sample.

**Figure 2 cancers-14-02933-f002:**
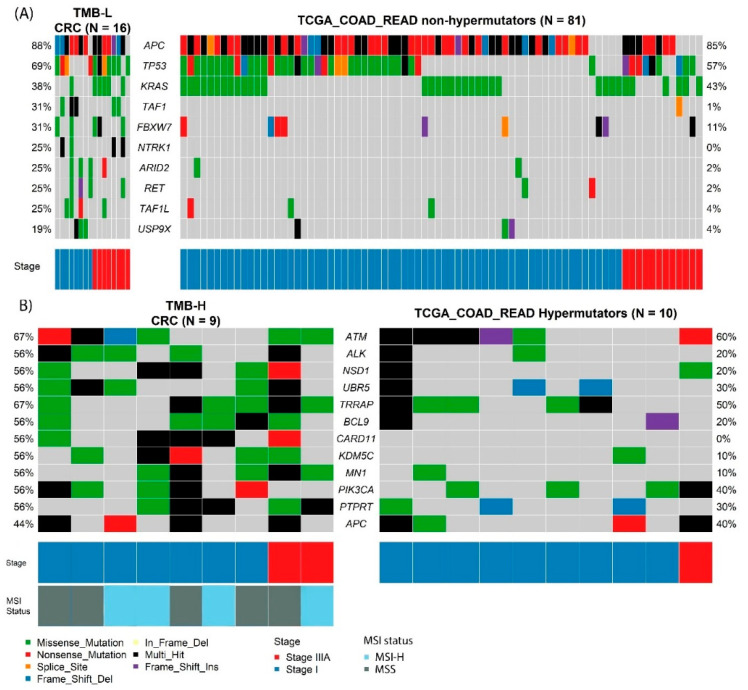
Co-Oncoplot of Stage I and IIIA CRC. (**A**) Top ten mutated genes in TMB-Low CRC and nonhypermutators TCGA COAD-READ cohort specimens. (**B**) Differentially distributed genes in TMB-High compared with TMB-Low and their distribution in TCGA COAD-READ hypermutator group.

**Figure 3 cancers-14-02933-f003:**
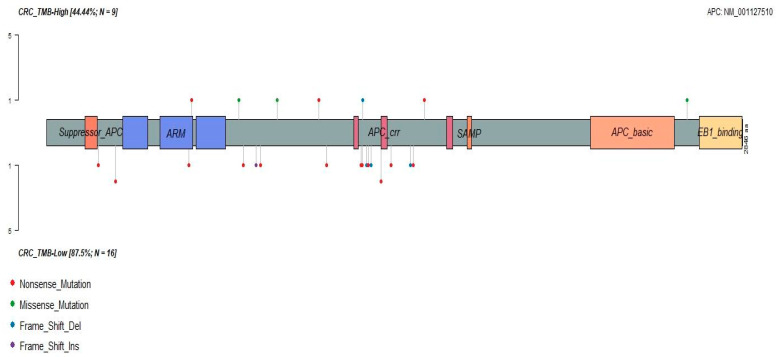
Lollipop plot shows distributions of *APC* variants in TMB-High and TMB-Low CRC. *APC* variants were present in 44% of TMB-H CRC cases distributed from codon 516–2620. In TMB-Low group, *APC* has a mutation frequency of 88%, and 58% of these variants are located in the mutation cluster region (MCR, codons 1282–1581) in the central part of *APC*.

**Figure 4 cancers-14-02933-f004:**
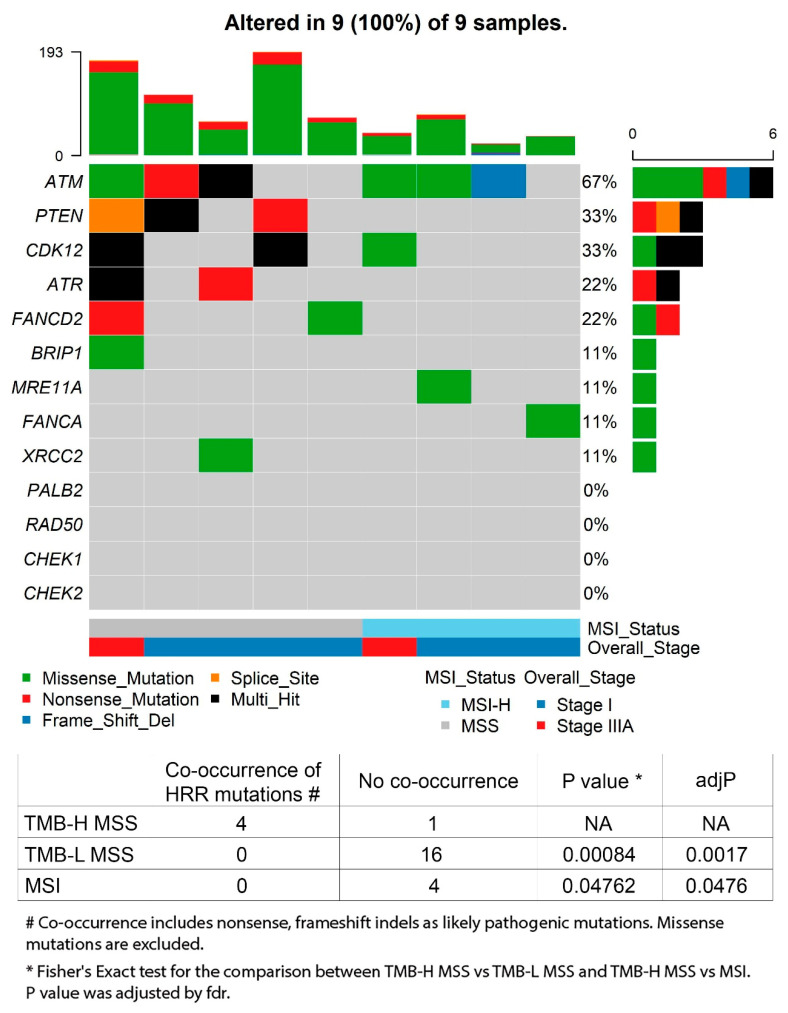
HRR gene co-occurrence in TMB-H MSS CRC.

**Figure 5 cancers-14-02933-f005:**
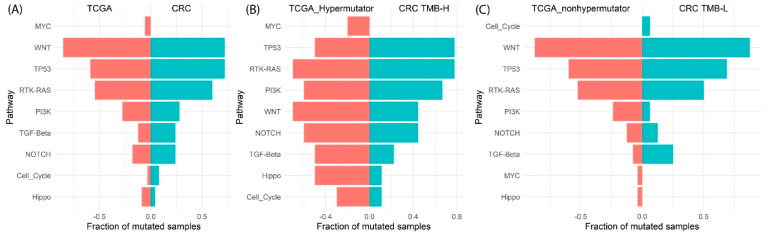
Pathway enrichment analysis in (**A**) TCGA−COAD−READ and CRC specimens, (**B**) TMB-H group compared with TCGA−COAD−READ hypermutator group, (**C**)TMB-L group compared with TCGA−COAD−READ nonhypermutated cohort.

**Table 1 cancers-14-02933-t001:** Clinical characteristics of Stage I and Stage IIIA CRC samples.

	Stage I (*n* = 16)	Stage IIIA (*n* = 10)	*p*-Value *
**Year**			
Range	2007–2017	2008–2018	
**Sex**			*p* = 0.692
Female	5 (31.25%)	4 (40.00%)	
Male	11 (68.75%)	6 (60.00%)	
**Age**			** *p* ** **= 0.038**
Range	49–87	31–85	
Mean(sd)	71.31 (11.07)	57.40 (21.36)	
**Tumour size**			*p* = 0.065
Range	15–30	13–55	
Mean(sd)	22.56 (4.91)	30.50 (15.36)	
**Site**			*p* = 0.806
Right	6 (37.50%)	3 (30.00%)	
Left	5 (31.25%)	2 (20.00%)	
Rectal	5 (31.25%)	5 (50.00%)	
**Total nodes**			*p* = 0.091
Range	13–22	12–49	
Mean(sd)	17.31 (3.11)	23.50 (13.66)	
**pT.7th.Ed**			** *p* ** **< 0.001**
T1	16 (100.00%)	6 (60.00%)	
T2	0 (0.00%)	4 (40.00%)	
**pN.7th.Ed**			
N0	16 (100.00%)	0 (0.00%)	By definition
N1a	0 (0.00%)	5 (50.00%)	
N1b	0 (0.00%)	3 (30.00%)	
N1c	0 (0.00%)	2 (20.00%)	
**Thin walled vessel invasion**			*p* = 0.055
0 (Absent)	15 (93.75%)	6 (60.00%)	
1 (Present)	1 (6.25%)	4 (40.00%)	
**Combined.BRAF.MMR.status**			*p* = 1.000
BRAF V600E +/MSS	1 (6.25%)	1 (10.00%)	
BRAF V600E −/MSH	0 (0.00%)	0 (0.00%)	
BRAF V600E +/MSH	3 (18.75%)	1 (10.00%)	
BRAF V600E −/MSS	12 (75.00%)	8 (80.00%)	

* Statistic methods: Student’s *t*-test for Age, Tumour size and Total nodes; Fisher’s exact test for Sex, Site, pN.7th.Ed, Thin walled vessel invasion and Combined BRAF/MMR status; *p* < 0.05 in bold.

**Table 2 cancers-14-02933-t002:** Statistical analysis * on differential distributed genes between Stage IIIA and Stage I.

Gene Symbol	Stage IIIA (freq%)	Stage I (freq%)	Pval	Odds Ratio	Confidence Interval. Upper	Confidence Interval. Lower	AdjPval
KRAS	5 (62.5%)	1 (12.5%)	0.12	9.76	625.22	0.68	0.24
TP53	7 (87.5%)	4 (50%)	0.28	6.16	391.63	0.42	0.38
APC	7 (87.5%)	7 (87.5%)	1	1	89.51	0.01	1

* Performed Fisher exact test on gene frequencies of two groups; *p* value was adjusted by fdr.

**Table 3 cancers-14-02933-t003:** Different distribution of common cancer genes in Stage I and IIIA hypermutated and nonhypermutated CRC.

Gene	TMB-H (*n* = 9)	TMB-L (*n* = 16)	Pval	AdjPval	TCGA COAD/READ Hypermutated (*n* = 10)	TCGA COAD/READ Nonhypermutated (*n* = 81)	Pval	AdjPval
ATM	67%	0%	**0.0005**	0.0090	60%	9%	**0.0004**	0.0006
ALK	56%	0%	**0.0024**	0.0090	20%	2%	0.0583	0.1486
NSD1	56%	0%	**0.0024**	0.0090	20%	2%	0.0583	0.1486
UBR5	56%	0%	**0.0024**	0.0090	30%	0%	**0.0010**	0.0114
TRRAP	67%	13%	**0.0099**	0.0188	50%	2%	**0.0001**	0.0002
BCL9	56%	6%	**0.0119**	0.0188	20%	1%	**0.0310**	0.0850
CARD11	56%	6%	**0.0119**	0.0188	0%	9%	1.0000	1.0000
KDM5C	56%	6%	**0.0119**	0.0188	10%	1%	0.2088	0.2491
MN1	56%	6%	**0.0119**	0.0188	10%	0%	0.1099	0.1513
PIK3CA	56%	6%	**0.0119**	0.0188	40%	20%	0.2178	0.2341
PTPRT	56%	6%	**0.0119**	0.0188	30%	6%	**0.0405**	0.0475
APC	44%	88%	0.0581	0.0736	40%	85%	**0.0035**	0.0048

Bold signifies significance with sufficient sample size for Fisher’s exact test using α 0.05 and β 0.8.

## Data Availability

Sequencing data is available from the NIH BioProject Accession PRJNA807367.

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
