# Peer review of "Deep Sequencing of Early T Stage Colorectal Cancers Reveals Disruption of Homologous Recombination Repair in Microsatellite Stable Tumours with High Mutational Burdens"

_cancers, 2022, doi:10.3390/cancers14122933_

Round 1

Reviewer 1 Report

The authors characterize the genetic features of Stage IIIA CRCs by targeted sequencing of cancer genes. Moreover, the authors compared the results to a similar analysis performed in a Stage I CRC cohort.

As pointed out in the manuscript, Stage IIIA CRC is rare and underrepresented in global genetic profiling efforts, such as TCGA. The authors analyzed 10 novel stage IIIA CRCs samples in this study. 

It's interesting to notice that Stage IIIA CRC patients were younger. 

I found this study overall interesting as filling the gap in terms of representation of specific stages of CRC is helpful. However, I would like to raise a few points:

1) The authors carefully mined TCGA, but this is not the only cohort of CRCs genetically profiled nowadays. Could the authors expand, expand or validate their findings by mining additional public datasets like GENIE or the Hartwig database, etc..? is stage not available for the samples included in these datasets?

2) overall, I believe that classifying samples in TMB-H or L can be misleading. In fact, maybe some observations the authors made could be explained by the different numbers of MSI and MSS samples within these two groups. I believe MSI status should be highlighted more in some of the figures and reported in the percentages of mutation frequency, at least in a supplementary file.

3) it's a bit unclear to me why the mutational load is so high in MSS samples with respect to non-MSI samples in this cohort. Any additional comments on that? If we believe in 'adenoma-carcinoma pathway' as a model of tumour evolution and tumours evolve through the different stages of CRC, how do you explain the high mutational load detected in this MSS cohort with respect to MSI samples, compared to more advanced stages? 

Author Response

1) The authors carefully mined TCGA, but this is not the only cohort of CRCs genetically profiled nowadays. Could the authors expand, expand or validate their findings by mining additional public datasets like GENIE or the Hartwig database, etc..? is stage not available for the samples included in these datasets?

Reply: We explored publicly accessible databases through the NIH GDC Data Portal, which already includes databases TCGA, TARGET, GENIE, CMI, BEATAML1.0, CGCI, CPTAC, CTSP, EXCEPTIONAL_RESPONDERS, FM (Foundation Medicine), HCMI, MMRF, MP2PRT, NCICCR, OHSU, ORGANOID, REBC, TRIO, VAREPOP, and WCDT. We applied filters for “primary tumor site” is “Colon” or “Rectum”, and “AJCC Pathologic Stage” is “stage i” or “stage iiia”. The remaining available projects are only TCGA, CPTAC and HCMI. The TCGA provides the largest number of samples which we used, CPTAC has only three stage IIIA cases while HCMI has only one stage IIIA available. We also investigated the Hartwig database, unfortunately they don’t have stage and tumor sub-type information available. Thus the use of TCGA dataset is the most appropriate.

2) overall, I believe that classifying samples in TMB-H or L can be misleading. In fact, maybe some observations the authors made could be explained by the different numbers of MSI and MSS samples within these two groups. I believe MSI status should be highlighted more in some of the figures and reported in the percentages of mutation frequency, at least in a supplementary file.

Reply: We don’t believe this is the case. We discuss more about these 2 groups on p6 last paragraph. All 4 MSI-H cases were grouped into the TMB-H cancers, so no MSI-H cancers are in the TMB-L group. MSI status is clearly indicated in Fig 1, Fig 2, Fig 4, Fig S3. MSI status is not relevant in other Figures.

3) it's a bit unclear to me why the mutational load is so high in MSS samples with respect to non-MSI samples in this cohort. Any additional comments on that? If we believe in 'adenoma-carcinoma pathway' as a model of tumour evolution and tumours evolve through the different stages of CRC, how do you explain the high mutational load detected in this MSS cohort with respect to MSI samples, compared to more advanced stages? 

Reply: The observation of higher TMB in some MSS specimens is very interesting and a key finding of our paper. We propose that mutations in homologous recombination DNA repair pathway explains the elevated TMB in these MSS samples (Fig 4, Table3). Tumors with instability for >20% of the four markers (MSH2, MSH6, MLH1, PMS2) are considered as MSI. It needs to be noted that MSI represents a tendency to mismatch repair and therefore a tendency to high TMB. Stage 1 and 3A are both low stage colon cancers (effectively stage 3A is ‘lower stage’, at least biologically than stage 2A,2B,2C as these patients have better prognoses), so as we are only looking at early stage disease it is not surprising that the MSI cancers have lower TMB than typically seen in advanced disease.

Reviewer 2 Report

Li et al. present an interesting manuscript where they aim to characterize molecular features of early T stage colorectal tumors.  They examine molecular differences between Stage I and Stage IIIA cases, where the later are defined by lymph node metastasis in spite of the primary lesion having limited invasive depth.  They also compare molecular features in TMB-H vs. TMB-L tumors using the combined set of Stage I and Stage IIIA cases.  In both cases, they use data from TCGA to illustrate the reproducibility of their results.

The manuscript addresses an important question from the translational point of view.  However, the work has important limitations.

Major concerns:

(1)   As the authors explicitly acknowledge at several points in the manuscript, the study is severely limited by small sample sizes, since the set of tumors that they sequenced includes only 16 Stage I and 10 Stage IIIA tumors.  This is a problem with two major implications:

(1.a) Some of the reported negative results (such as those shown in Table 2) might be due to a lack of statistical power, particularly for cases where the magnitude of the difference in frequency is very large (e.g., KRAS frequencies of 62.5% vs. 12.5%).  The use of TCGA data for comparison purposes partially ameliorates this situation, but the small number of Stage IIIA specimens in the TCGA set remains a source of concern.  To better understand this issue, I think that it would be beneficial to perform a power analysis and to get an accurate estimate of the magnitude of the difference that the study would be able to detect at a statistically significant level given the sample size used.  Ideally, this will be provided separately for the set of 26 samples generated by the authors and also for the larger set of TCGA samples that they used.  

(1.b) A rigorous, systematic approach to statistical testing and multiple hypotheses correction is needed in order to ensure that the reported results are not just observed by chance.  However, no p-values are provided for the results shown in Table 3.  Were these the only genes tested for differences in the TMB-H vs. TMB-L groups? If so, why? If not, what was the full list of genes that were tested and what was the p-value associated to each one, both unadjusted and adjusted?  Since TMB-H tumors have more mutations, it is not surprising to find many genes that are mutated more often in that group... so if the authors want to focus on ATM, they need to explain and justify this choice better.

(1.c) Also, are ATM mutations really occurring more often that would be expected by chance in the TMB-H group? Or could this just be explained by a combination of the higher background mutation rate within this set and the relatively high length of the gene? Would ATM be significant in a MutSigCV (or MuSic) analysis restricted to TMB-H tumors from the samples sequenced in the manuscript? What about the same analysis using the cohort from TCGA?

(2) The choice of the mean TMB (16.5 mut/Mb) as a cut-off to separate TMB-H from TMB-L seems arbitrary and is not well justified. Why was this choice made instead of using the median, or the third quartile, or the 90th percentile?  In Figure 1 of the TCGA-COADREAD manuscript [PMID:22810696],  the cut-off selected by TCGA seems reasonable because we see a clear gap in the TMB measured along the vertical axis between hypermutated and non-hypermutated tumors.  However, no such gap is clearly observed at 16.5 mut/Mb in Figure 1 of the current manuscript. If anything, one could argue that the biggest gaps are rather observed at 75 mut/Mb, or 50 mut/Mb? This choice of cut-off value is very important for the manuscript, because the whole comparison of TMB-H vs. TMB-L tumors depends strongly on it.

Additional concerns:

  • To ensure reproducibility, the authors should provide a list of TCGA sample identifiers for the TCGA cases that were used in the analysis, including detailed staging information.
  • “A matched cohort of 16 Stage I CRC was assembled for comparison”. How was this “matching” done? Were the 16 stage I cases chosen at random and happened to have not strong clinical differences with respect to the Stage IIIA cases? Or were they chosen in order to avoid such differences?  This needs to be clearly explained in the text.

  • In Figure 3 and overall description of APC differences: language is speculative and claims about specific differences and enrichments need to be supported by statistical test results, otherwise observations could be due to chance (particularly given the small sample size).

  • Are any of the co-occurrence patterns reported in Figure 4 statistically significant? Given the high TMB in this set, it might not be surprising to find several mutated genes within a given sample. Can the authors provide some measure of statistical significance? Maybe use some valid test or run some type of random permutation analysis?

  • Did the authors filter out variants of unknown significance for the results reported in Figure 5? In the reference mentioned in the text (PMID: 29625050), the authors provided a detailed list of alterations as part of their curated pathway templates and only included mutations that were considered to have oncogenic potential based on annotations from the OncoKB database. In the absence of this detailed filtering and curation, the frequencies reported in Figure 5 may well be over-inflated, particularly in the panel for the hypermutated/TMB-H cases (for example, it seems very suspicious to see 100% of cases with Hippo pathway alterations).

  • Did any of the TMB-H cases had POLE mutations? This is an important question and it is not currently discussed in the text.

I believe that all the concerns detailed above should be properly addressed before the manuscript can be accepted for publication.

Author Response

(1.a) Some of the reported negative results (such as those shown in Table 2) might be due to a lack of statistical power, particularly for cases where the magnitude of the difference in frequency is very large (e.g., KRAS frequencies of 62.5% vs. 12.5%).  The use of TCGA data for comparison purposes partially ameliorates this situation, but the small number of Stage IIIA specimens in the TCGA set remains a source of concern.  To better understand this issue, I think that it would be beneficial to perform a power analysis and to get an accurate estimate of the magnitude of the difference that the study would be able to detect at a statistically significant level given the sample size used.  Ideally, this will be provided separately for the set of 26 samples generated by the authors and also for the larger set of TCGA samples that they used.  

Reply: We are very clear about the limitation of sample size, and this reflects the rarity of Stage IIIA CRC. As noted to Reviewer 1, TCGA is the largest publicly available dataset for Stage IIIA CRC, so is the most useful comparator which mostly ameliorates this issue. The reviewer’s points are interesting and we performed power analysis as suggested to include significant findings in an updated Table 3. We focus on robust findings validated with the TCGA dataset.

(1.b) A rigorous, systematic approach to statistical testing and multiple hypotheses correction is needed in order to ensure that the reported results are not just observed by chance.  However, no p-values are provided for the results shown in Table 3.  Were these the only genes tested for differences in the TMB-H vs. TMB-L groups? If so, why? If not, what was the full list of genes that were tested and what was the p-value associated to each one, both unadjusted and adjusted?  Since TMB-H tumors have more mutations, it is not surprising to find many genes that are mutated more often in that group... so if the authors want to focus on ATM, they need to explain and justify this choice better.

Reply: We have modified Table 3 by including those cancer genes that have a minimum of 5 mutational events in our dataset, rather than including a long gene table with fewer events. Shown in Table 3 are the Fisher’s exact test with pvalue and adjusted pvalue with sufficient sample size for validity.

(1.c) Also, are ATM mutations really occurring more often that would be expected by chance in the TMB-H group? Or could this just be explained by a combination of the higher background mutation rate within this set and the relatively high length of the gene? Would ATM be significant in a MutSigCV (or MuSic) analysis restricted to TMB-H tumors from the samples sequenced in the manuscript? What about the same analysis using the cohort from TCGA?

Reply: We ran the MutSigCV analysis on our TMB-H group and also on TCGA hypermutated dataset as requested. Nothing is significant in both datasets, which we assume is due to small sample sizes. We include a new explanatory note to acknowledge that we cannot exclude the possibility that ATM mutation could be due to higher background mutation rate in our and TCGA sample set.

(2) The choice of the mean TMB (16.5 mut/Mb) as a cut-off to separate TMB-H from TMB-L seems arbitrary and is not well justified. Why was this choice made instead of using the median, or the third quartile, or the 90th percentile?  In Figure 1 of the TCGA-COADREAD manuscript [PMID:22810696],  the cut-off selected by TCGA seems reasonable because we see a clear gap in the TMB measured along the vertical axis between hypermutated and non-hypermutated tumors.  However, no such gap is clearly observed at 16.5 mut/Mb in Figure 1 of the current manuscript. If anything, one could argue that the biggest gaps are rather observed at 75 mut/Mb, or 50 mut/Mb? This choice of cut-off value is very important for the manuscript, because the whole comparison of TMB-H vs. TMB-L tumors depends strongly on it.

Reply: We observed that the top 5 cases with highest TMB were MSS genotypes. We wanted to investigate what was different in these from the other MSS genotypes with significantly lower TMB. These two groups of MSS genotypes are clearly different as measured by TMB median (45 vs 4.4) or TMB mean (49 vs 5.9). To allow comparison with TCGA hypermutator groups (both MSI-H and MSS) we included the 4 MSI-H cancers creating a group of 9 cancers with elevated TMB that are statistically different from the remaining 16 samples which we termed TMB low (Ttest P<0.0001).

Additional concerns:

  • To ensure reproducibility, the authors should provide a list of TCGA sample identifiers for the TCGA cases that were used in the analysis, including detailed staging information.

Reply: We provide MAF file used in this study as a new Supplementary table 3.

  • “A matched cohort of 16 Stage I CRC was assembled for comparison”. How was this “matching” done? Were the 16 stage I cases chosen at random and happened to have not strong clinical differences with respect to the Stage IIIA cases? Or were they chosen in order to avoid such differences?  This needs to be clearly explained in the text.

Reply: We have edited the manuscript (p6) to report that the 16 stage 1 cases were randomly selected, matched with the Stage IIIa for gender and tumour site and had sufficient quality DNA extracted from FFPE to enable deep sequencing. They reflect the typical clinicopathological features of stage 1 CRC seen in our institution. 

  • In Figure 3 and overall description of APC differences: language is speculative and claims about specific differences and enrichments need to be supported by statistical test results, otherwise observations could be due to chance (particularly given the small sample size).

Reply: We have changed the Figure 3 legend to simply state where the APC mutations are distributed.

  • Are any of the co-occurrence patterns reported in Figure 4 statistically significant? Given the high TMB in this set, it might not be surprising to find several mutated genes within a given sample. Can the authors provide some measure of statistical significance? Maybe use some valid test or run some type of random permutation analysis?
    Reply: Fisher’s exact test was used to demonstrate significance of co-occurrence mutations, shown in Fig 4.

  • Did the authors filter out variants of unknown significance for the results reported in Figure 5? In the reference mentioned in the text (PMID: 29625050), the authors provided a detailed list of alterations as part of their curated pathway templates and only included mutations that were considered to have oncogenic potential based on annotations from the OncoKB database. In the absence of this detailed filtering and curation, the frequencies reported in Figure 5 may well be over-inflated, particularly in the panel for the hypermutated/TMB-H cases (for example, it seems very suspicious to see 100% of cases with Hippo pathway alterations).
    Reply: Thankyou for the comment. We have reanalysed this data following removal of VUS following the method in the TCGA paper. We present a revised Fig 5, and limit our observations to report the high frequency mutations observed in our dataset compared with TCGA data analysed in the same manner.

  • Did any of the TMB-H cases had POLE mutations? This is an important question and it is not currently discussed in the text.

Reply: We have included a statement on pg6 that POLE is not included in the Ampliseq panel so could not be determined. We find it unlikely given POLE incidence in CRC is very low (~1%) and we do not see ultra-high mutated specimens.

Round 2

Reviewer 1 Report

I have no further questions or major concerns. I think the revised manuscript is significantly improved. 

Author Response

The reviewer had no further questions

Reviewer 2 Report

I appreciate the effort made by the authors to add p-values to Table 3, the fact that they run MutSigCV and the fact that they rerun their pathway analyses after filtering out variants of unknown significance.  I think that the quality of the manuscript has improved and the text can be accepted if the following minor concerns are properly addressed.

I am still a bit concerned about the lack of a proper analysis of statistical power. Even though the authors claim that they have “performed power analysis as suggested”, I do not see this type of analysis anywhere in the manuscript.  Just to be clear, by performing a power analysis I do not mean just computing p-values, but instead getting an estimate of the magnitude of the difference in biological signals that could be detected with a certain level of confidence based on the available sample size.  For example: “Assuming a KRAS mutation rate of X% and a 2-sided type 1 error of 5%, we will be able to detect an odds ratio (OR) of Y with 80% power in a two-sample test of proportions”.  Still, since their negative results are reproduced using TCGA, I think that this is not a strong requirement for publication.  In any case, I would suggest to at least include a sentence to explicitly mention that the lack of association might be just a consequence of strong limited statistical power due to the small sample sizes.

In the absence of additional clarification, I find the following statement problematic: “The ultra-hyper-mutator gene POLE was not part of the AmpliSeq panel, but we find it unlikely to be a driver in our dataset given we did not observe cases with ultra-high mutational loads.” What is considered an “ultra-high mutational load” for the AmpliSeq panel? The top 2 MSS cases with the highest TMB in Figure 1 have TMBs that are three times higher than the most mutated MSI cases; to me this suggests that these are likely POLE hypermutators. Otherwise, what biological mechanism could explain such as high TMB?  This is consistent with Fig. 1 from the COADREAD TCGA manuscript [PMID: 22810696], where non-MSI tumors with the mutation rates above the higher mutation rate for MSI cases were all POLE.  Since it seems that POLE status cannot be reliably determined from the AmpliSeq panel, I think that the authors should clearly mention this as a limitation of the study and remove or rephrase their current claim of not having any hypermutated cases.  They should also exercise caution at other points of the manuscript where they talk about having "made interesting observations on mutational burden and pathway defects in high mutation burden MSS CRC that has not been widely reported" (and similar statements), because these observations might be in fact driven by undetected POLE hypermutants.

It seems to me from reading the text that no matched normal specimens (tissue or blood) were used to filter out germline mutations.  Is this correct? If so, this should be explicitly stated.  This could also provide a potential explanation for the higher TMB observed in the study when compared to TCGA (where matched normals had been used) and this might be worth mentioning together with the current explanation from the authors: “Furthermore, our detection of more variants is likely attributed to the robustness of sensitive deep sequencing method we used, compared with WXS used in TCGA projects.“

Minor typo in the discussion: “Lee et al predicted the best performed three individual genes were PTEN, ATM and ATR” -> best performing

Author Response

I appreciate the effort made by the authors to add p-values to Table 3, the fact that they run MutSigCV and the fact that they rerun their pathway analyses after filtering out variants of unknown significance.  I think that the quality of the manuscript has improved and the text can be accepted if the following minor concerns are properly addressed.

I am still a bit concerned about the lack of a proper analysis of statistical power. Even though the authors claim that they have “performed power analysis as suggested”, I do not see this type of analysis anywhere in the manuscript.  Just to be clear, by performing a power analysis I do not mean just computing p-values, but instead getting an estimate of the magnitude of the difference in biological signals that could be detected with a certain level of confidence based on the available sample size.  For example: “Assuming a KRAS mutation rate of X% and a 2-sided type 1 error of 5%, we will be able to detect an odds ratio (OR) of Y with 80% power in a two-sample test of proportions”.  Still, since their negative results are reproduced using TCGA, I think that this is not a strong requirement for publication.  In any case, I would suggest to at least include a sentence to explicitly mention that the lack of association might be just a consequence of strong limited statistical power due to the small sample sizes.

Reply: We thank the reviewer for articulating the structure for power analysis. We present these findings in Table S4, which confirmed the reported genes, apart from TRRAP, which we deleted. We applied the same power analysis to the TCGA cohort and could confirm ATM and UBR5. We have added a sentence pg.8 as suggested to mention the limited associations are likely due to small sample sizes of TMB-H and TCGA hypermutated samples.

In the absence of additional clarification, I find the following statement problematic: “The ultra-hyper-mutator gene POLE was not part of the AmpliSeq panel, but we find it unlikely to be a driver in our dataset given we did not observe cases with ultra-high mutational loads.” What is considered an “ultra-high mutational load” for the AmpliSeq panel? The top 2 MSS cases with the highest TMB in Figure 1 have TMBs that are three times higher than the most mutated MSI cases; to me this suggests that these are likely POLE hypermutators. Otherwise, what biological mechanism could explain such as high TMB?  This is consistent with Fig. 1 from the COADREAD TCGA manuscript [PMID: 22810696], where non-MSI tumors with the mutation rates above the higher mutation rate for MSI cases were all POLE.  Since it seems that POLE status cannot be reliably determined from the AmpliSeq panel, I think that the authors should clearly mention this as a limitation of the study and remove or rephrase their current claim of not having any hypermutated cases.  They should also exercise caution at other points of the manuscript where they talk about having "made interesting observations on mutational burden and pathway defects in high mutation burden MSS CRC that has not been widely reported" (and similar statements), because these observations might be in fact driven by undetected POLE hypermutants.

Reply: We have looked closely at our findings in reference to literature. We point out the large study by Campbell et al. published in Cell (PMID: 29056344) which analysed ~78,000 adult tumours and defined POLE mutated phenotypes as ultra-hypermutators with TMB exceeding 100/Mb. This definition is becoming more common in the literature. Campbell et al. further showed that hypermutated sample TMB spanning 10-100/Mb range included MSI and MSS specimens, which is entirely consistent with our grouping. Consistent with our findings in early T stage CRC, Lee et al.    (PMID: 32352724) reported MSS hypermutated specimens without POLE pathogenic exonuclease domain mutations were characterised by pathogenic mutations in some HRR genes. These studies demonstrate that hypermutated MSS CRC do not necessarily require pathogenic POLE variants. We have included this explanation on pg7 and also a sentence to state that the absence of POLE on the Ampliseq panel prevented us from definitely excluding POLE from these samples.

It seems to me from reading the text that no matched normal specimens (tissue or blood) were used to filter out germline mutations.  Is this correct? If so, this should be explicitly stated.  This could also provide a potential explanation for the higher TMB observed in the study when compared to TCGA (where matched normals had been used) and this might be worth mentioning together with the current explanation from the authors: “Furthermore, our detection of more variants is likely attributed to the robustness of sensitive deep sequencing method we used, compared with WXS used in TCGA projects.“

Reply: We modified the method to indicate that filtering was based on public germline data as no matched normals were used. Further, we elaborated on this to contrast the study designs by modifying the identified sentence on pg8.

Minor typo in the discussion: “Lee et al predicted the best performed three individual genes were PTEN, ATM and ATR” -> best performing

Reply: thanks, fixed.